# Polyethylene Terephthalate Hydrolases in Human Gut Microbiota and Their Implications for Human Health

**DOI:** 10.3390/microorganisms12010138

**Published:** 2024-01-10

**Authors:** Heqi Zhou, Songbiao Shi, Qiuhong You, Kaikai Zhang, Yuchuan Chen, Dekai Zheng, Jian Sun

**Affiliations:** 1State Key Laboratory of Organ Failure Research, Key Laboratory of Infectious Diseases Research in South China, Ministry of Education, Guangdong Provincial Key Laboratory of Viral Hepatitis Research, Guangdong Provincial Clinical Research Center for Viral Hepatitis, Department of Infectious Diseases, Nanfang Hospital, Southern Medical University, Guangzhou 510515, China; zzuheqizhou@163.com (H.Z.); youqiuhong38@smu.edu.cn (Q.Y.); kairzhang2021@126.com (K.Z.); 18407349746@163.com (Y.C.); zdk19940206@foxmail.com (D.Z.); 2South China Sea Institute of Oceanology, Chinese Academy of Sciences, Guangzhou 510301, China; sbshi@scsio.ac.cn

**Keywords:** polyethylene terephthalate, human gut microbiota, PET hydrolases, micro-plastics, cytokine expression

## Abstract

Polyethylene terephthalate (PET), primarily utilized for food and beverage packaging, consistently finds its way into the human gut, thereby exerting adverse effects on human health. PET hydrolases, critical for the degradation of PET, have been predominantly sourced from environmental microbial communities. Given the fact that the human gut harbors a vast and intricate consortium of microorganisms, inquiry into the presence of potential PET hydrolases within the human gut microbiota becomes imperative. In this investigation, we meticulously screened 22,156 homologous sequences that could potentially encode PET hydrolases using the hidden Markov model (HMM) paradigm, drawing from 4984 cultivated genomes of healthy human gut bacteria. Subsequently, we methodically validated the hydrolytic efficacy of five selected candidate PET hydrolases on both PET films and powders composed of micro-plastics (MPs). Notably, our study also unveiled the influence of both diverse PET MP powders and their resultant hydrolysates on the modulation of cytokine expression in macrophages. In summary, our research underscores the ubiquitous prevalence and considerable potential of the human gut microbiota in PET hydrolysis. Furthermore, our study significantly contributes to the holistic evaluation of the potential health hazards posed by PET MPs to human well-being.

## 1. Introduction

Humans are continually exposed to micro-plastics (MPs) due to the pervasive presence of plastic items in everyday life. The negative impacts of MPs are currently under intensive investigation, and related studies have expanded rapidly [1,2,3,4]. MPs have been well-established as risk factors for both the environment and organisms. This field of research has garnered attention not only from environmental science and oceanography but also from the medical field. MPs primarily encompass fibers, particles, or thin films with sizes <5 mm. This category includes polyethylene (PE), polypropylene (PP), polyvinyl chloride (PVC), polyamide (PA), polyethylene terephthalate (PET), and polystyrene (PS). Notably, PET, widely used in food packaging and water bottles [5], stands out as one of the most prevalent MPs found in the human gastrointestinal tract [6]. Quantitative analysis revealed that PET MPs were present in adult fecal samples at a median concentration of 5800 ng/g. This concentration was significantly higher, by an order of magnitude, in infant fecal samples, with a median concentration of 83,000 ng/g [7]. As such, the significance of considering and evaluating the hazards posed by PET MPs cannot be overstated.

PET hydrolases, which catalyze the breakdown of PET polymer into mono(2-hydroxyethyl) terephthalic acid (MHET), terephthalic acid (TPA), and ethylene glycol (EG), enabling the degradation and recycling of PET waste, have primarily been isolated from microorganisms. Notably, *Is*PETase from *Ideonella sakaiensis* 201-F6 has been identified as one such PET hydrolase [8]. Given the extensive microbial community, predominantly comprising bacteria, present in the human gut, concerns have arisen about the potential bidirectional relationship between ingested MPs and the human microbiota. The presence of PET MPs in human stool samples has confirmed the direct interaction between PET MPs and gut microorganisms [6]. Recent studies have demonstrated that the consumption of PET MPs affects the composition and diversity of colonic microbial communities [9]. Despite these advances, our understanding of the gastrointestinal fate and hazards posed by ingested PET MPs remains limited. Notably, only Tamargo et al. have reported a significant deposition of colonic microbiota on the surface of PET MPs following dynamic *in vitro* colonic fermentations [9]. Biofilm formation has been regarded as the initial step in the degradation of plastics. These aforementioned pieces of evidence imply a potential link between the biodegradation of PET MPs and the human gut microbiota. Currently, there is a lack of research into the potential of the human gut microbiota to degrade PET.

TPA represents the primary hydrolysis product of PET. Although data on the toxicity of TPA in animals indicate low toxicity, researchers have found that chronic, long-term TPA exposure induces bladder epithelial cell proliferation and an increased risk of stone formation, particularly in weanling rats [10,11]. Moreover, evidence from Luciani-Torres et al. suggests that TPA activates DNA damage markers and pathways in human breast epithelial cells [12]. Molonia et al. have also observed that even nanomolar concentrations of TPA affect adipocyte thermogenesis and induce the NF-κB proinflammatory pathway [13]. As TPA is considered a structural analog of phthalates, which are endocrine-disrupting chemicals, a reevaluation of safe levels of TPA exposure has been proposed.

Based on the current literature, we found that the research on PET hydrolases is more focused on genetic code expansion to realize energy recycling with high efficiencies and low costs. Few studies have focused on the potential of the human gut microbiota for PET degradation and the effects of PET hydrolysates on human health. We screened potential PET hydrolases from databases containing genomes of healthy human gut bacteria, verified their hydrolytic activity and explored their impacts on cell viability and cytokine expression levels in macrophages. Our findings underscore the capacity of the human gut microbiota to depolymerize PET and emphasize the importance of considering the harms associated with PET MPs in terms of both the plastic particles themselves and their resulting hydrolysates.

## 2. Materials and Methods

### 2.1. Chemicals and Substrates

Our analyses utilized two types of PET sourced from Dongguan Zhangmutou, China: (i) polydisperse PET powders (with a crystallinity >25% and particle size <400 μm), and (ii) amorphous PET film (with a crystallinity of 7% and a thickness of 250 μm). Terephthalic acid (TPA) (CAS: 100-20-1, product code: P816020-100 g), mono(2-hydroxyethyl) terephthalic acid (MHET) (CAS: 1137-99-1, product code: H909109-100 mg), and dimethyl sulfoxide (DMSO) (CAS: 67-68-5, product code: D806645-500 mL) were procured from Macklin, Shanghai, China. TPA and MHET were dissolved in DMSO, and the medium containing the same concentration of DMSO was used as a vehicle control. Macrophage colony-stimulating factor (M-CSF) (product code: HY-P7085), lipopolysaccharides (LPS) (product code: HY-D1056), and murine interleukin-4 (IL-4) (product code: HY-P70644) were obtained from MedChemExpress, Princeton, NJ, USA. 

### 2.2. Databases and Bioinformatic Analysis Employed in the Study

The reference dataset consisted of 74 sequences of PET hydrolases derived from natural diversity [14]. We obtained 1520 reference genomes of cultivated healthy human gut bacteria from the Culturable Genome Reference (CGR) study (https://www.ncbi.nlm.nih.gov/bioproject/PRJNA482748/, accessed on 1 June 2023) [15], and an additional 3464 genomes from the Broad Institute-Open Biome Microbiome Library (BIO-ML) study (https://www.ncbi.nlm.nih.gov/bioproject/PRJNA544527, accessed on 1 June 2023) [16]. To identify homologous sequences, we employed both the HMM webpage (http://hmmer.org, accessed on 5 June 2023) and a local instance of the software (v3.1b2) to search within the pool of 4984 reference genomes using the 74 PET hydrolase sequences as queries. For the construction of phylogenetic trees, the neighbor-joining method was implemented in the MEGA 11 program, with a bootstrap value of 1000 resampling replicates to enhance robustness [17]. To predict protein structures, Alpha Fold2 was utilized with default parameters, while the forecast of active pockets was conducted through PASSer (https://passer.smu.edu, accessed on 15 July 2023). Structural visualization and alignments were executed using PyMOL (v2.0), while the calculation of electrostatic potential employed APBS (ver. 3.0.0) via the APBS web service. 

### 2.3. Protein Expression and Purification

To explore the potential PET-degrading enzymes, five candidate genes, HG1-5, were codon-optimized and custom-synthesized by Tianyi Huayu Bioscience (Wuhan, China), and subsequently integrated into a pET-21a vector for expression in *Escherichia coli* BL21(DE3). Detailed sequence information can be found in Appendix A. The bacterial cultures were aerobically cultivated in lysogeny broth (LB) medium supplemented with 100 μg/mL ampicillin at 37 °C until reaching an OD600 of 0.6–0.8. At this point, isopropyl *β*-d-1-thiogalactopyranoside (IPTG) was introduced to attain a final concentration of 0.6 mM, followed by further incubation at 16 °C for 18 h. The harvested cells were subjected to centrifugation (4 °C, 8000× *g*, 5 min) and subsequently lysed via ultrasonication using buffer A (500 mM NaH_2_PO_4_, 100 mM NaCl, 30 mM imidazole). Subsequently, the proteins, which were equipped with a C-terminal histidine tag, underwent purification through nickel-ion affinity chromatography employing Ni-NTA agarose (Sangon Biotech, Shanghai, China). Elution was achieved using buffer B (500 mM NaH_2_PO_4_, 100 mM NaCl, 300 mM imidazole). The purified protein fractions were subjected to dialysis against storage buffer C (500 mM NaH_2_PO_4_, 100 mM NaCl, pH 8.0). The assessment of protein concentrations and evaluation of purity were carried out using 12%-SDS-PAGE analysis.

### 2.4. Enzyme Activity Assays

The determination of PET-hydrolytic activity was conducted using previously established methods [18]. In a concise overview, 15 μg of purified enzyme was incubated with amorphous PET films (8 mm in diameter, 15 mg, crystallinity of 7%) in a 0.5 mL buffer containing 100 mM K_2_HPO_4_ (pH = 7.0). The mixture was then subjected to heating at 37 °C using a thermocycler for 24 h. To terminate the reaction, we diluted the reaction solution ten times and treated it in boiling water for 10 min. The reaction products after centrifugation were analyzed using high-performance liquid chromatography (HPLC) with an Agilent 1200 system (Santa Clara, CA, USA), featuring an SB C-18 column (5 μm, 4.6 × 150 mm; Agilent). The HPLC conditions involved UV detection at 260 nm and a linear gradient of methanol ranging from 10% to 100%, with a flow rate of 0.6 mL/min. The resulting degradation products, namely MHET and TPA, were quantified. The hydrolysis of PET MP powders was performed following the same aforementioned method.

### 2.5. Scanning Electron Microscopy (SEM) 

In order to examine the surfaces of PET powders and films, a scanning electron microscope (HITACHI, Tokyo, Japan) was employed. This was carried out under high-vacuum conditions and with a variable high tension ranging from 0.3 to 30 kV. Prior to observation, the plastic films and powders underwent washing and drying processes. Furthermore, a thin, 10-nanometer layer of gold was sputter-coated onto them to ensure adequate electrical conductivity.

### 2.6. Cell Culture

In the case of immortalized cell lines, we acquired the human intestinal cell line Caco2 and the colon epithelial cell line FHC, alongside murine monocyte macrophages RAW264.7 cells, from the American Type Culture Collection (ATCC, Rockefeller, MD, USA). The FHC and RAW 264.7 cells were nurtured in Dulbecco’s Modified Eagle Medium (DMEM) (Gibco, Carlsbad, CA, USA), supplemented with 10% fetal bovine serum (FBS) and 1% penicillin/streptomycin. As for the Caco2 cell line, it was cultured in RPMI 1640 medium (Gibco, Carlsbad, CA, USA), augmented with 10% FBS and 1% penicillin/streptomycin.

Regarding primary cells, mouse bone-marrow-derived macrophages (BMDMs) were procured and employed in our investigation. The procedures for this have been thoroughly detailed in a prior publication [19]. To provide a concise overview, BMDMs were isolated from the femurs and tibias of mice, followed by cultivation in RPMI 1640 medium supplemented with 10% FBS, 1% penicillin/streptomycin, and 20 ng/mL of macrophage colony-stimulating factor (M-CSF). After a maturation period of 7 days, the BMDMs were ready for subsequent experiments. All cell cultures were maintained at a temperature of 37 °C with a 5% CO_2_ atmosphere. The protocols involving animals were approved by the Institutional Animal Care and Use Committee of Nanfang Hospital, Southern Medical University (Guangzhou, China).

### 2.7. Cell Treatment

To assess cell viability, we employed the Cell Counting Kit-8 (CCK8) test. In this study, a total of 3000 cells were seeded in 96-well plates with 100 μL of complete medium. Following 48 h of exposure to varying concentrations of TPA (0.01, 0.10, 0.50, and 1.00 mM), MHET (1.00 mM), or polydisperse PET MPs (0.01, 0.10, 0.50, and 1.00 mg/mL), we measured the absorbance at 450 nm using a universal microplate reader. This measurement served as an indicator of the number of viable cells.

For RNA extraction assays, RAW 264.7 cells or differentiated matured BMDMs were cultured in complete medium containing TPA (0.01, 0.10, 0.50, and 1.00 mM), MHET (1.00 mM), or polydisperse PET MPs (0.01, 0.10, 0.50, and 1.00 mg/mL) overnight. Subsequently, these cells were stimulated with 50 ng/mL LPS to induce macrophage M1 polarization or 20 ng/mL IL-4 for M2 activation for 6 h. Prior to use, all BMDMs underwent the aforementioned 7-day differentiation process. Following stimulation, the culture medium was aspirated, and the relevant cells were rinsed with pre-chilled phosphate-buffered saline before being harvested using a scraper.

### 2.8. Real-Time Polymerase Chain Reaction

Cell samples were collected, and the EZ-press RNA purification Kit (product code: B0004D, EZBioscience, Roseville, CA, USA) was employed to extract cellular RNA. Subsequently, the RNA underwent reverse transcription into cDNA using the 4× Reverse Transcription Master Mix (code # A0010GQ, EZBioscience, Roseville, California, USA), following the manufacturer’s instructions. The resulting cDNA (5 ng/mL) was then employed for quantitative polymerase chain reaction, utilizing the 2× SYBR^®^ Green Pro Taq HS Premix (code # AG11701; ACCURATE BIOLOGY, Changsha, China), performed on the Roche LightCycler480 (Basel, Switzerland). The relative expression levels of genes were determined via the 2-∆∆CT method with GAPDH as the reference gene. Specific gene primers can be found in Appendix A. 

### 2.9. Statistical Analysis 

The results are presented as means ± standard deviation. Graphical representation and statistical analysis were conducted using GraphPad Prism (version 8.0). For data comprising more than two groups with equal variance, a one-way analysis of variance (ANOVA) followed by Tukey’s test was employed for comparisons. Significance levels were denoted as follows: *, **, and *** represented *p* < 0.05, *p* < 0.01, and *p* < 0.001, respectively. *p* < 0.05 was considered significant. The absence of a label indicates the absence of a significant difference, unless explicitly stated otherwise.

## 3. Results

### 3.1. Exploration of PET Hydrolase Candidates in Human Gut Bacterial Genomes

In the pursuit of identifying PET hydrolases within the human gut microbiota, we embarked on a search for putative PET hydrolase sequences within a comprehensive collection of approximately 5000 reference genomes derived from cultivated human gut bacteria. This endeavor was accomplished through the utilization of an HMM model. The overall procedure is depicted in Figure 1A. Initially, we accessed a catalog encompassing 74 PET hydrolases from seven distinct phylogenetic groups, showcasing a spectrum of PET-degrading enzymes with natural diversity [14]. Subsequently, we employed the HMM approach to probe the vast genomic dataset, housing 4984 genomes of cultivated human gut bacteria. These genomes were sourced from both Chinese (1520 genomes) and Bostonian (3464 genomes) populations. Guided by criteria entailing over 60% coverage and 40% amino acid sequence similarity, our inquiry yielded a harvest of 22,156 homologous sequences. Notably, these sequences initially surfaced with notable HMM scores (>60%) and a compelling E-value threshold (<1.0 × 10^−10^). However, it is noteworthy that these homologous sequences were predominantly drawn from merely one to four distinct phylogenetic groups. This phenomenon could be attributed to the fact that the five to seven lineages were ultimately derived from actinomycetes. The selection of candidate PET hydrolases was based on the following process. (A) We meticulously selected the top 10 sequences from each of the four distinctive phylogenetic groups based on their HMM scores. (B) The resultant compilation of 40 amino acid sequences was harnessed for the construction of a neighbor-joining tree. Intriguingly, upon this analysis, it came to light that four sequences were redundant (Appendix A), thus necessitating their removal (A3-4, A3-5, A3-8, A4-1). (C) Ultimately, the remaining 36 candidates, along with 3 novel PET hydrolases (MG 8-10) retrieved from human sample metagenomes, as reported by Bhumrapee et al. [20], were integrated into a neighbor-joining tree alongside the previously documented 74 PET hydrolase sequences (Figure 1B) [14]. (D) The selection of these 36 sequences was grounded in their coherent clustering with groups one to four, standing apart from the previously reported PET hydrolase sequences of human microbiome origin. (E) To characterize the universality of the encompassing PET hydrolysis activity within the gut microbiota, factors including enzyme classification and bacterial species diversity were carefully considered. (F) This comprehensive evaluation led us to choose five potential PET hydrolases from the pool of thirty-six candidates. These five selected enzymes, namely HG1-5, were subsequently subjected to meticulous examination to affirm their PET-degrading potential (please refer to A1-2, A2-5, A2-10, A3-7, and A4-10).

### 3.2. PET-Degrading Activity Assessment

For the following experiments, enzymes HG1-5 were expressed and purified, with signal peptides removed. Remarkable production and purity of HG1-5 proteins were evident through SDS-PAGE analysis (Figure 1C). To assess the hydrolytic activity against PET, amorphous PET films, and MP powders (<400 μm) were employed as substrates, alongside *Is*PETase serving as a positive control. PET hydrolases catalyzed PET into mono(2-hydroxyethyl) terephthalic acid (MHET), terephthalic acid (TPA), and ethylene glycol (Figure 2A). Analysis of released aromatic monomers (comprising MHET and TPA) confirmed the presence of these compounds in HG2-5, excluding HG-1. This outcome underscores the effective PET hydrolysis capacity of these enzymes, yielding TPA/MHET in varying proportions over a 24h period at 37 °C. The exceptional hydrolysis activity exhibited by HG-3 was of particular note, which liberated 42 μM TPA within 24 h (Figure 2B). However, the impact of these five enzymes on high crystallinity MP powders was limited compared to amorphous PET films. No hydrolysates were detected within a 24-h incubation at 37 °C. Furthermore, the examination of PET film surfaces revealed distinct pits and grooves induced by HG1-5, unequivocally demonstrating the PET hydrolysis activity of all five enzymes (Figure 2C). The presence of discernible pits was confined primarily to HG-3 among these powders (Figure 2D), indicating distinct treatment responses between PET materials of varying crystallinity.

### 3.3. Characterization and Structural Prediction of Putative PET Hydrolases

Through multiple sequence alignment, we initially identified five enzymes belonging to the α/β hydrolase superfamily that harbor conserved residues, specifically Gly-X-Ser-X-Gly, as well as the catalytic triad Ser-His-Asp (Appendix A). Our sequence analysis categorized these enzymes primarily within *Proteobacteria* and *Firmicutes*. They fall under the classifications of carboxylesterase, ferulic acid esterase, and esterase/lipase (Table 1). Notably, ferulic acid esterase and esterase/lipase are prevalent among gut microbiota, including common inhabitants like *Lactobacillus*, *Holdemanella*, and *Bacteroides* [21,22]. This underscores that gut bacteria possess the capability to degrade PET plastics. Employing Alpha Fold 2, we conducted a structural analysis of the five selected enzymes and positive control *Is*PETase. The five enzymes exhibited distinct structural folds and surface charge distribution with *Is*PETase (Figure 3). The PET hydrolase candidates displayed a similar core structure and catalytic triad compared with *Is*PETase. However, they contained different cap domains around the active center, which presented different active pockets (Figure 3A) and may hinder the binding between the active center and the substrate, thus affecting catalytic activity (Appendix A). A typical example is HG-1, where a larger lid domain leads to lower catalytic activity. The surface charges of five selected enzymes and *Is*PETase showed different surface charge characterizations (Figure 3B). The surface charge and catalytic activity of PET hydrolases have been reported to correlate, and a positive surface charge near the active center renders catalytic reactions more effective [20]. HG1-5 contain acidic surface charges, consistent with their predicted isoelectric points (pI 4.47–6.66, Table 1), which may also be one of the reasons why the catalytic activity was lower than the positive control *Is*PETase.

### 3.4. Effects of PET Microplastics (MPs) and Hydrolysates on Cell Viability and Macrophage Polarization

Given the widespread presence of PET hydrolases, we investigated the potential risks associated with hydrolysis products. Considering that ingested PET MPs initially encounter the human intestinal lumen, we explored the response of the human intestinal cell line Caco2 and the colon epithelial cell line FHC to MHET, TPA, and polydisperse PET MPs. Cell viability was assessed using CCK8 analysis, and the data demonstrated that TPA and MHET had no discernible impact on cell proliferation in either cell line, even at concentrations of up to 1.00 mM (Appendix A). Furthermore, we assessed the viability of the two cell lines when exposed to PET MPs in powder form. Both Caco2 and FHC cell lines were subjected to different concentrations (0.01, 0.10, 0.50, 1.00 mg/mL) of PET MP powders for 48 h, yet no notable difference in cell viability was observed when compared to the control group (Appendix A). The polydisperse and irregular PET MPs, with sizes <400 μm, were characterized using scanning electron microscopy, as depicted in Appendix A.

The presence of numerous immune cells within the intestinal mucosa is widely acknowledged, playing a pivotal role in maintaining intestinal homeostasis. Among these, intestinal macrophages, primarily originating from bone marrow monocytes, have a central role in the inflammatory response and interact with bacterial metabolites in the gut [23,24]. Activated macrophages generally adopt either a pro-inflammatory M1 phenotype or an anti-inflammatory M2 phenotype depending on the microenvironment. In vitro studies reveal that M1 macrophages can be induced by lipopolysaccharides (LPS), while interleukin-4 (IL-4) leads to the polarization of M2 macrophages, which produce relevant cytokines [25].

We first investigated whether TPA or MHET influenced macrophage polarization. As depicted in Figure 4A, TPA at 0.01 mM increased the expression of M1-associated genes IL-1β and IL-6, while reducing the expression of the M2-associated gene Arg-1. At 1.00 mM, TPA further elevated mRNA levels of IL-1β, IL-6, and TNF-α, and decreased Arg-1 expression (Figure 4B). Additionally, TPA exacerbated the mRNA levels of IL-1β, IL-6, and TNF-α induced by LPS, while dampening IL-4-stimulated Arg-1 in bone-marrow-derived macrophages (BMDMs) (Figure 4C). A similar trend was observed in RAW 264.7 cells treated with TPA (0.10 or 0.50 mM), although statistical significance was not achieved (Appendix A). Conversely, MHET up to 1.00 mM had no significant effect on cytokine expression in RAW264.7 cells (Appendix A).

Furthermore, when M1 macrophages were exposed to PET MP powders at concentrations of 0.01 or 0.10 mg/mL, the expression of M1-associated genes was upregulated, particularly IL-1β, while Arg-1 was downregulated. However, both IL-1β and Arg-1 were significantly downregulated following incubation with PET MPs at concentrations of 0.50 or 1.00 mg/mL (Figure 4D,E and Appendix A). A direct correlation between cell viability and cytokine production was evident. Appendix A illustrates the viability of RAW264.7 cells treated with PET MP powders, showing that relatively low concentrations of PET MPs had no notable effects on cell proliferation. However, treatment with PET MP powders at 0.50 and 1.00 mg/mL resulted in a degree of cell death, with viabilities of approximately 83.5% and 72.9%, respectively. Interestingly, without any stimulation, the expression of cytokines in the macrophages treated with PET MPs alone did not change, as shown in Figure 4F,G. 

## 4. Discussion

Numerous studies have highlighted the detrimental effects of ingested MPs on both humans and animals [4]. However, the gastrointestinal outcomes and risks associated with ingested MPs remain poorly comprehended. A recent investigation indicated that the enzymatic hydrolysis of polylactic acid MPs can generate potentially more toxic oligomers during gastrointestinal processes [26]. PET, a prevalent type of MPs found in human fecal matter, has been shown to undergo hydrolysis by various environmental microorganisms [8,27,28]. Presently, our understanding of the potential of the human gut microbiota to degrade PET and its consequent impact on human health is limited. This study aims to screen potential PET hydrolases based on databases containing genomes of healthy human gut bacteria. Subsequently, we verified their hydrolytic activity on PET films and MPs. Moreover, we observed that both polydisperse PET MPs and their hydrolysates affected the expression of cytokines in macrophages. Our work advances the comprehension of the potential health hazards posed by PET MPs to humans.

In earlier studies, the majority of novel enzyme discoveries related to PET breakdown were metagenome data-driven, particularly those sourced from marine environments [20,29]. More recently, Eiamthong et al. introduced new PET hydrolases, named MG8, from the metagenome of human saliva, with subsequent analyses focusing on enzyme genetic modification [20]. While metagenomics is effective and wide-ranging for identifying PET-degrading sequences, our interest lay in pinpointing PET hydrolase candidates and the associated cultivated microorganisms. We sought these candidates among cultivated healthy human-gut bacterial genomes, as these cultivable bacteria are conducive to subsequent mechanistic research. Studies have suggested that environmental changes may drive microbial alterations or evolution [20]. Given the increasing consumption of plastic products and exposure to ingested MPs in the human gut, investigating potential PET hydrolases from the human gut seems warranted. Utilizing an HMM model to screen 4984 cultivated human-gut bacterial genomes, we identified 22,156 homologous sequences, illustrating the substantial reservoir of PET hydrolase candidates in the gut microbiota. Through experimental validation, we confirmed that five distinct enzymes (HG1-5) from different microbial groups possessed PET hydrolysis activity. Genus such as *Enterobacter*, *Holdemanella*, and *Weizmannia* were found to exhibit relatively higher abundance in the human gut. These findings underscore the ubiquity and enormous potential of gut microbiota in PET hydrolysis. However, the catalytic activity of HG1-5 was much lower than that of PET hydrolase *Is*PETase from environmental microorganisms. It was found that the size of the active pocket and surface charge can affect the catalytic activity [20]. The positive charge around the active center of the hydrolase is more conducive to substrate adsorption. Hydrolase HG-1 is a typical example, in which the larger cap domain forces the substrate to be difficult to contact with the active center, resulting in lower PET catalytic activity. These factors that affect catalytic activity also provide a good direction for the subsequent enzyme modification. The PET hydrolase discovered from gut microbiota provides excellent resources for the subsequent clearance of PET MPs in the intestine, and further exploration is needed for the applied biotechnology.

In assessing the impact of MPs on human health, considering chemical risk alongside particle toxicity and gut microbiota perturbation was a natural approach. Detailed description of the gut microbiota perturbation caused by PS MPs was presented in our previous work [30,31]. In addition, prior studies have reported a synergistic toxic effect between MPs and substances like bisphenol A and phthalates absorbed by MPs, leading to intensified intestinal inflammation and metabolic disorders [32,33]. Ingested PET MPs might come into contact with intestinal epithelial cells, particularly under pathological conditions such as inflammatory bowel disease, where the intestinal mucosa barrier function is compromised [34]. We found that microscale PET MPs had no significant impact on the viability of Caco2 and FHC cells, aligning with previously reported data [35]. Additionally, we observed no effects on cell proliferation from MHET and TPA, suggesting either their lack of toxicity toward fundamental cellular activities or only minimal toxicity.

Intestinal macrophages are well known for maintaining intestinal homeostasis, but their polarization and cytokine production responses to PET MPs are not fully understood. We discovered that cytokine expression in macrophages treated solely with PET MPs remained unchanged, yet pro-inflammatory cytokines increased and anti-inflammatory cytokines decreased following LPS/IL-4 stimulation. These findings qualitatively aligned with previous data involving 25–200 μm PP MPs applied to the same cell line [36]. This suggested that PET MPs may exacerbate immune response to pathogens instead of triggering immune response alone. However, in our study, PET MPs at a relatively high concentration downregulated both M1 and M2 markers, which might relate to diminished cell activity. Notably, the irregular shape of the PET MPs that we employed may play a role. Choi et al. suggested that irregular-shaped PE MPs with sharp edges and higher curvature differences correlated with immune cell cytotoxicity [37]. These findings reinforce the notion that the impact of MPs on macrophage inflammatory response is determined by the size, concentration, and surface properties rather than material composition. Furthermore, in addition to the adverse effects mentioned above, TPA, a primary hydrolysate, propelled macrophages toward a pro-inflammatory phenotype. Our findings align with the existing literature, as TPA was shown to activate the NF-κB pathway [13], a significant regulator of M1 macrophage polarization [25]. Collectively, our data demonstrate that both PET MPs and their hydrolysates, including TPA, can exacerbate macrophage inflammatory responses, posing potential health risks to humans.

## 5. Conclusions

In summary, our study delved into cultivated data from healthy human gut bacterial genomes, leading to the identification of five PET candidate hydrolases: HG1-5. We successfully confirmed the hydrolytic activity of HG1-5 on PET films and that of HG-3 on both PET films and PET MPs. Furthermore, we shed light on the widespread presence and immense potential of the human gut microbiota in PET degradation. Additionally, our investigation revealed that both polydisperse PET MPs and PET hydrolysates exerted an influence on cytokine expression in vitro within macrophages. This underscores the necessity of considering the detrimental effects of PET MPs in terms of both the plastic material itself and its resultant hydrolysates. Our findings significantly contribute to a comprehensive assessment of the potential health risks posed by PET MPs to humans.

## Figures and Tables

**Figure 1 microorganisms-12-00138-f001:**
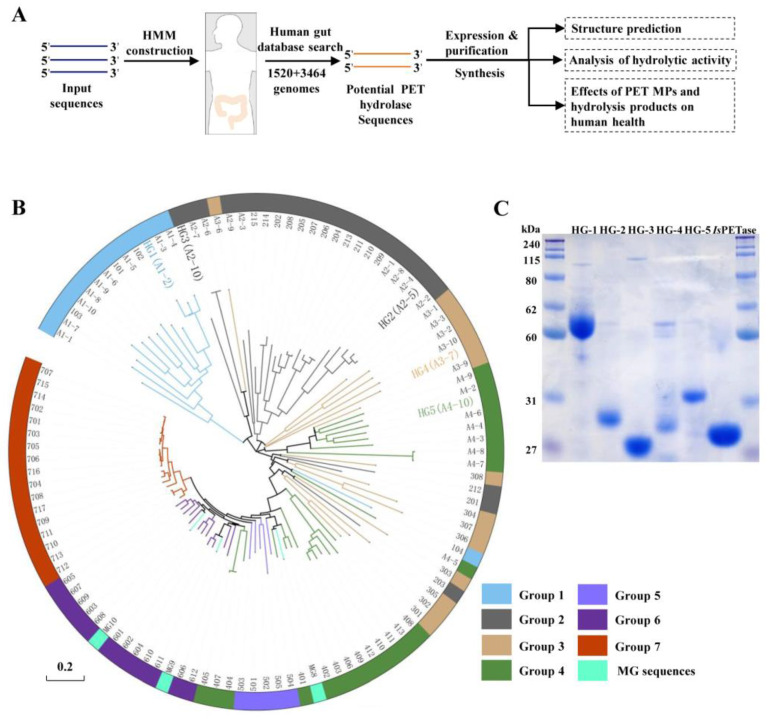
**Discovering PET hydrolase candidates from human gut bacteria genomes.** (**A**) Workflow used in this study to identify putative PET hydrolases from human gut bacteria whole genomes through a hidden Markov model (HMM) model, using known PET-degrading sequences as a query. Structure prediction and functional experiments were further performed on five putative PET hydrolases. (**B**) Neighbor-joining tree of 36 candidates, 74 reference PET hydrolases, and 3 PET hydrolases from metagenome of human samples (MG 8-10). Groups 1 to 7 represent seven previously reported distinct phylogenetic groups of PET hydrolases from nature diversity [14]. (**C**) SDS-PAGE of purified HG1-5 and *Is*PETase.

**Figure 2 microorganisms-12-00138-f002:**
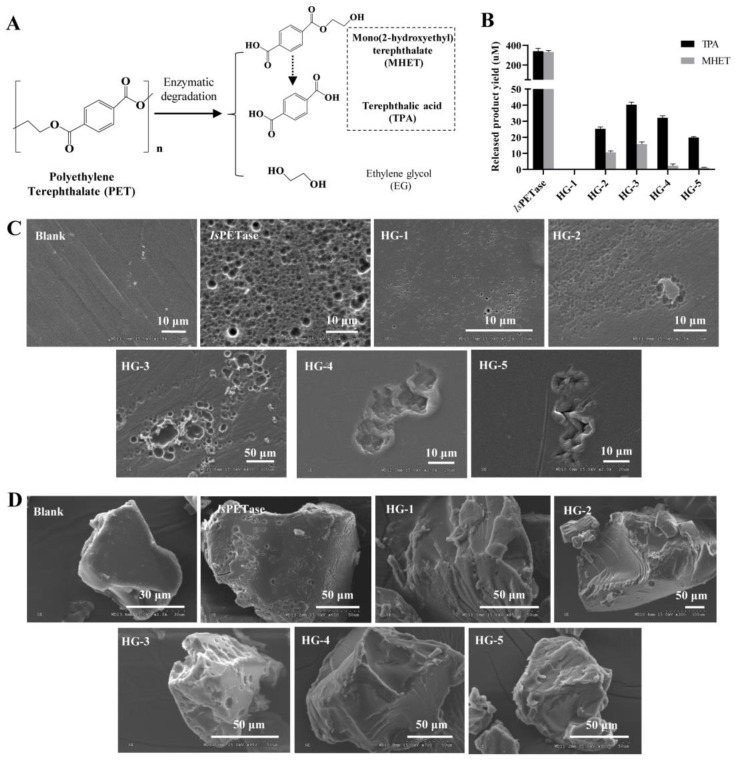
**Identification of PET-degrading activities on PET plastic.** (**A**) PET hydrolases catalyzed PET into mono(2-hydroxyethyl) terephthalic acid (MHET), terephthalic acid (TPA), and ethylene glycol. (**B**) Hydrolysis products released from the PET films after incubation with purified HG1-5 proteins at 37 °C (pH = 7.0) for 24 h. *Is*PETase was used as positive control. (**C**,**D**) Scanning electron microscopy (SEM) of amorphous PET films (**C**) and PET micro-plastic (MP) powders (**D**) incubated with purified HG1-5 proteins at 37 °C (pH = 7.0) for 24 h. *Is*PETase was used as positive control. All SEM images were taken after 24 h of incubation with 15 μg purified protein in 0.5 mL buffer containing 100 mM K_2_HPO_4_ (pH = 7.0) or in a buffer-only control.

**Figure 3 microorganisms-12-00138-f003:**
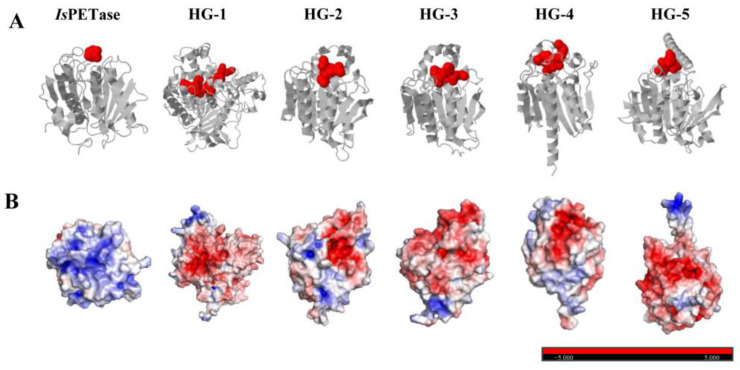
**Surface charge and activate site pocket of five PET hydrolase candidates and *Is*PETase.** (**A**) Predicted three-dimensional structure of HG1-5 and *Is*PETase, with activate site pocket shown as red spheres. (**B**) Surface charge characterization of HG1-5 and *Is*PETase, with red and blue colors representing regions of positive and negative charge, and white color representing uncharged. Red and blue colors represent negative (acidic residues) and positive (basic residues) potentials, respectively (scale of 5.0 to +5.0 kBT/ec).

**Figure 4 microorganisms-12-00138-f004:**
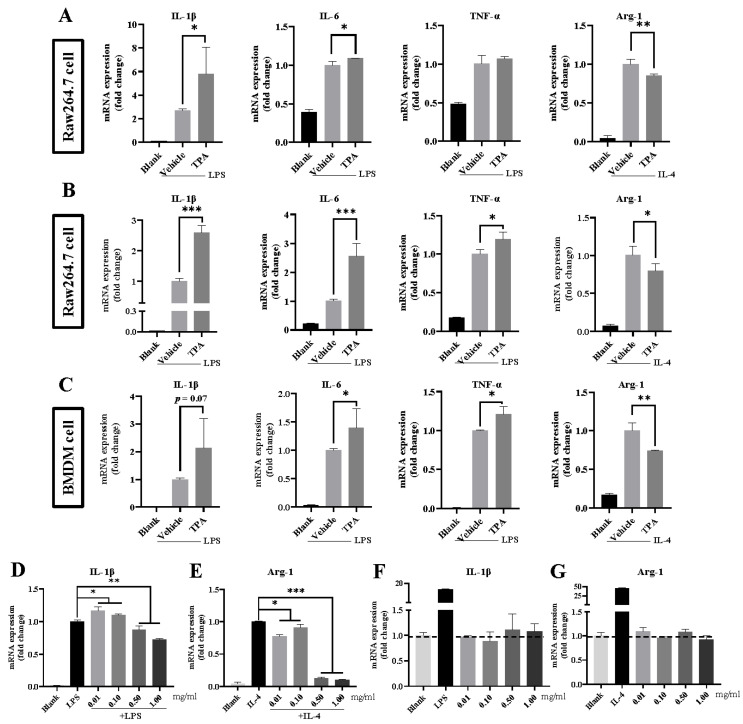
**Effects of PET MPs and hydrolysates on cell viability and macrophage polarization.** (**A**,**B**) Relative expression levels of IL-1β, IL-6, TNF-α, and Arg-1 (IL-4 stimulation) in mouse monocyte macrophages RAW 264.7 cells treated with TPA (0.01 mM) (**A**) and TPA (1.00 mM) (**B**) for 24 h followed by LPS/IL-4 stimulation for 6 h. (**C**) Relative expression levels of IL-1β, IL-6, TNF-α, and Arg-1 (IL-4 stimulation) in mouse bone-marrow-derived macrophages (BMDMs) treated with TPA (1.00 mM) for 24 h followed by LPS/IL-4 stimulation for 6 h. (**D**,**E**) Relative expression levels of IL-1β (**D**) and Arg-1 (**E**) in RAW 264.7 cells treated with PET MPs (<400 μm) at a gradient dose at 0.01, 0.10, 0.50, and 1.00 mg/mL for 24 h followed by LPS/IL-4 stimulation for 6 h. (**F**,**G**) Relative expression levels of IL-1β (**F**) and Arg-1 (**G**) in RAW 264.7 cells treated with PET MPs (<400 μm) at a gradient dose at 0.01, 0.10, 0.50, and 1.00 mg/mL for 24 h without any stimulation. *** *p*-value < 0.001; ** *p*-value < 0.01; * *p*-value < 0.05.

**Table 1 microorganisms-12-00138-t001:** Characteristics of expressed proteins HG1-5.

Sample Number	Accession Number	Protein Molecular Weight (kDa)	Protein Isoelectric Point	Family	Species
HG-1	WP_047344220.1	55.39	6.66	Carboxylesterase	*Enterobacter* sp. AM17-1
HG-2	WP_118011433.1	29.17	6.16	Ferulic acid esterase	*Holdemanella biformis*
HG-3	RGD93181.1	29.79	4.47	Ferulic acid esterase	*Clostridiales bacterium* AM23-16LB
HG-4	WP_013858543.1	28.42	6.44	Esterase/lipase	*Weizmannia coagulans*
HG-5	WP_008690040.1	35.08	4.52	Alpha/beta hydrolases	*Longicatena* sp.

## Data Availability

The genomes of health human gut bacteria are available from the NCBI-BioProject (accession PRJNA482748 and PRJNA544527). All data are available from the corresponding author on request.

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
