# Peer review of "Polyethylene Terephthalate Hydrolases in Human Gut Microbiota and Their Implications for Human Health"

_microorganisms, 2024, doi:10.3390/microorganisms12010138_

Round 1

Reviewer 1 Report

Comments and Suggestions for Authors

Zhou et al. have investigated the presence of PET hydrolases within the gut microbiome using bioinformatic searches followed by cloning and enzyme characterization. This revealed several putative esterases that had activity against PET films, though not micro-plastic particles. These particles and their hydrolysis particles also exhibited pro-inflammatory effects in cell culture models. Together this is a well conducted study that helps advance our understanding of the impacts of micro-plastics.

Section 2.4 – how long were the activity assays incubated for and was there any means of stopping the reaction employed?

Figure 1B – what are the groups that are colored in the phylogenetic tree? This should be either included in the figure itself or at least in the legend.

Figure 2 and legend appears to be incorrect. The way the figure is laid out, the HPLC results should be panel B and the SEM panels C and D. However, the HPLC results are not even mentioned in the legend. There is also not enough information in this figure legend for the SEM images, such as incubation time, amount of enzyme added, scale of the images.  

Table 1 and Figure S2 have a discrepancy as to the identity of HG-5. The table indicates that it is in the Longicatena genus, while the figure indicates that it is from Turicibacter sanguinis. Please correct whichever of these is incorrect.

Section 3.3 please change the title from ‘Structural Determination’ to ‘Structural Prediction’

Figure 3A What function is being served by highlighting the positive and negatively charged regions? Isn’t the substrate neutral? The products I suppose would be negatively charged, but are you suggesting any relevance to interactions between these products and the enzymes? It is not mentioned at all in the text and I am not sure what the relevance would be.

Figure 3B What exactly are these red spheres, the putative catalytic triad? 1) this needs to be better described and 2) it would probably be more useful to replace the charge plot in a) with a zoomed in view of the proposed active site showing the side chains of the proposed catalytic residues and surrounding amino acids to give an idea of the kinds of substrates that could fit in there. Also, it would be good to show what the positive control used in these experiments looks like.

Together this might give some insight into why HG-1 has no detectable activity in the HPLC assay, which should be commented upon. Also, the fact that HG-1 has a different fold and is twice the size of the other enzymes should be commented upon more. Is this larger size a consistent feature in the other enzymes in its section of the tree?

Line 432-433 “We successfully confirmed their hydrolytic activity on both PET films and PET MPs” But no activity was detected on the PET MPs. Please revise

Author Response

Please see the attachment,thank you.

Reviewer 2 Report

Comments and Suggestions for Authors

Generally speaking I like the subject and the way the paper was presented.

I suggest the following regarding the data:

1.authors need to change figure 1C which does not correspond with the table of molecular weights of proteins. protein number HG-3 is the most different from the table

2. groups analyzed in figure 1 need to be better explained in the figure captions 

minor thing.

In the table the authors need to write "Protein molecular WEIGHT"

Comments on the Quality of English Language

I think that some of the expressions used by the authors need to be avoided:

any study is supposed to be meticulous and supposedly remarkable

Author Response

Please see the attachment,thank you.

Reviewer 3 Report

Comments and Suggestions for Authors

The article does not contain serious flaws and required some minor revisions:

1. Italicize microbial taxa.

2. Line 239. “This comprehensive evaluation led us to choose five potential PET hydrolases from the pool of 36 candidates.” Please, briefly describe the criteria for selection of PET hydrolases from the pool of the sequences analyzed.

3. Line 248. “IsPETase serving as a positive control (Fig. 2A)”. Fig. 2A does not contain any results regarding positive control. Could you include data of positive control variant in Fig. 2?

4. Line 249. “distinct pits and grooves induced by …. all five enzymes” and “aromatic monomers ….excluding HG-1.”. Could explain the absence of monomers in the case of the treatment with HG-1 (despite pits revealed).

5. “3.5. Figures” section. It is a bit of strange section. Remove separate section and distribute figures throughout the text.

Author Response

Please see the attachment,thank you.

Round 2

Reviewer 1 Report

Comments and Suggestions for Authors

All comments have been addressed